# Metastatic Kidney Cancer: Does the Location of the Metastases Matter? Moving towards Personalized Therapy for Metastatic Renal Cell Carcinoma

**DOI:** 10.3390/biomedicines12051111

**Published:** 2024-05-16

**Authors:** Catalin Baston, Andreea Ioana Parosanu, Ioana-Miruna Stanciu, Cornelia Nitipir

**Affiliations:** 1Faculty of Medicine, Carol Davila University of Medicine and Pharmacy, 8 Sanitary Heroes Boulevard, 050474 Bucharest, Romania; catalin.baston@umfcd.ro (C.B.); ioana-miruna.stanciu@drd.umfcd.ro (I.-M.S.); cornelia.nitipir@umfcd.ro (C.N.); 2Department of Urology, Fundeni Clinical Institute, 022328 Bucharest, Romania; 3Department of Oncology, Elias University Emergency Hospital, 011461 Bucharest, Romania

**Keywords:** renal cell carcinoma, metastatic sites, immunotherapy, targeted therapy

## Abstract

The management of renal cell carcinoma (RCC) has been revolutionized over the past two decades with several practice-changing treatments. Treatment for RCC often requires a multimodal approach: Local treatment, such as surgery or ablation, is typically recommended for patients with localized tumors, while stage IV cancers often require both local and systemic therapy. The treatment of advanced RCC heavily relies on immunotherapy and targeted therapy, which are highly contingent upon histological subtypes. Despite years of research on biomarkers for RCC, the standard of care is to choose systemic therapy based on the risk profile according to the International Metastatic RCC Database Consortium and Memorial Sloan Kettering Cancer Centre models. However, many questions still need to be answered. Should we consider metastatic sites when deciding on treatment options for metastatic RCC? How do we choose between dual immunotherapy and combinations of immunotherapy and tyrosine kinase inhibitors? This review article aims to answer these unresolved questions surrounding the concept of personalized medicine.

## 1. Introduction

Worldwide, kidney cancer represents 5% of all cancers in men and 3% in women. According to GLOBOCAN 2020 data, RCC is the 14th most commonly diagnosed cancer and accounts for over 85% of all primary renal neoplasms [1,2,3].

The incidental detection of kidney tumors has significantly increased due to the widespread use of radiologic imaging. However, survival is heavily influenced by the stage of diagnosis. Approximately one-third of patients with RCC have metastatic disease, with a five-year survival rate of only 12% [4,5,6,7].

Kidney cancer can spread to any part of the body and present itself in various ways. Nevertheless, lung metastasis accounts for 45–80% of cases, followed by bone (25–35%), lymph node (20–25%), and liver (18–20%). Less frequently, 4–11% of patients experience brain metastases from RCC. Additionally, there are specific, unusual, and challenging sites for RCC metastases, such as the pancreas, duodenum, or thyroid [8,9,10,11].

These different sites of metastases have prognostic significance. For instance, patients with lung-only metastases have higher survival rates than those with metastases to other sites. On the other hand, patients with metastases to the liver, bones, and brain tend to have the worst prognosis. However, patients with endocrine metastases, such as pancreatic metastases, even though rare, exhibit a significantly prolonged disease course and a better outcome [12,13,14,15].

Furthermore, the metastatic sites influence how we treat patients. For brain and bone metastases, we may combine radiotherapy with systemic treatment or, when possible, consider surgical intervention. In some cases, particularly for liver metastases where local therapy options may be limited, we must explore more aggressive treatments, including combination therapies [16,17,18].

Due to the lack of consensus on the relationship between kidney metastasis location and therapeutic response, we reviewed the current landscape of metastatic RCC therapy, with a focus on its effectiveness in specific metastatic locations.

## 2. Trends of Systemic Therapy against Metastatic RCC

The therapy landscape for RCC has shifted dramatically over the last four decades. Whenever possible, cytoreductive nephrectomy is the primary treatment for RCC. In select patients with early tumors, active surveillance or tumor ablation are also viable alternatives. However, systemic therapy continues to serve as the backbone of advanced RCC treatment [19,20]. 

Historically, kidney cancer has been considered a radiation-resistant tumor. Despite significant progress and challenges in kidney cancer treatment, RCC remains essentially resistant to chemotherapy [21,22,23].

Various approaches have been developed to treat metastatic renal cell carcinoma, with a focus on clear cell carcinoma, the most common histological subtype. Immunotherapy has been the cornerstone of RCC treatment for over 20 years due to its high immunogenicity [24]. Cytokine-based therapy using high-dose interleukin-2 and interferon alfa was the mainstay of systemic treatment from the 1980s to 2005. In the early 2000s, the treatment landscape for metastatic RCC underwent a significant shift with the introduction of targeted therapy using growth factor tyrosine kinase inhibitors (TKIs). However, immune checkpoint inhibitors (ICIs) have revolutionized the treatment of metastatic RCC, demonstrating durable tumor remission and long-term safety [25,26].

Today, ICI-based combination therapies, such as doublet ICIs or combinations of ICIs with TKIs, have become the standard of care in metastatic RCC. The rationale behind this strategy is to target the two hallmarks of RCC: the immunogenic tumor microenvironment and angiogenesis [27,28]. 

According to current guidelines, there are five immune-based (IO)-combinations approved to optimize clinical outcomes in RCC patients: ipilimumab-plus-nivolumab, avelumab + axitinib, axitinib–pembrolizumab, lenvatinib–pembrolizumab, and cabozantinib–nivolumab. Researchers compared these IO combinations with sunitinib and demonstrated longer progression-free survival or overall survival [29]. Unfortunately, the combination of atezolizumab-plus-bevacizumab, evaluated in the phase III IMmotion151 trial, did not reveal a significant improvement in overall survival compared to sunitinib in previously untreated patients with metastatic RCC. Therefore, this regimen is excluded from the current guidelines [30].

### 2.1. IO/IO or IO/TKI. What Is the Difference?

Each regimen has different mechanisms of action and specific adverse events, making it crucial to provide patient-centered care considering individual comorbidities. Immunotherapy has revolutionized cancer treatment, but many trials excluded patients with active or pre-existing autoimmune disorders [31]. In the CheckMate 214 clinical trial, 93% of patients who were treated with nivolumab and ipilimumab experienced some form of immune-related adverse reactions. These adverse reactions included endocrine, pulmonary, hepatic, renal, digestive, or skin issues, and 46% of these cases were considered grade three or four events. Additionally, up to 35% of patients required high doses of glucocorticoids to treat these immune-related adverse reactions. [32]. In contrast, the combination of IO-TKI was associated with a decreased incidence of irAEs. For instance, in the Javalin 101 trial, 38.2% of patients had adverse events categorized as immune-related, with only 9.0% being grade three or higher [33]. Consequently, IO/TKI combinations are preferred for patients with severe autoimmune conditions [34,35]. On the other hand, high blood pressure is a class effect of vascular endothelial growth factor tyrosine kinase inhibitors. Therefore, hypertensive patients may be candidates for single-agent IO or IO-IO combination therapy [36,37].

There are several critical differences between these treatment regimens. A breakthrough with IO-IO therapy is its potential to achieve durable responses. Immunotherapy can provide long-term immune memory, enabling the immune system to continually adapt and potentially offer longer-lasting remissions [38,39,40].

Nevertheless, IO/TKI combinations present several other advantages. Despite their undesirable effects, such as hypertension, impaired wound healing, or proteinuria associated with TKIs, it is essential to highlight that the objective response rate consistently tends to be higher with IO-TKI therapies [41,42]. Primary progression is up to 10% in IO-TKI, compared to 20% with IO-IO [43]. Furthermore, in settings with bulky, high-volume RCC, IO/TKI combinations prove to be especially valuable. Lastly, another advantage of the IO/TKI combination is the observed survival benefit across all metastatic renal cell carcinoma (mRCC) risk groups [44,45]. 

### 2.2. Are All Regimens Equally Effective?

As a benchmark, the CheckMate 214 trial established the outstanding efficacy of nivolumab-plus-ipilimumab over sunitinib in patients with metastatic RCC. Extended follow-up in the five-year analysis confirmed durable efficacy benefits for first-line nivolumab-plus-ipilimumab compared with sunitinib. CheckMate 214 allowed enrolment of all International Metastatic RCC Database Consortium (IMDC) risk categories. Nevertheless, among favorable-risk patients, the response rate was higher with sunitinib. Therefore, nivolumab-plus-ipilimumab has become the standard recommended regimen for intermediate- and poor-risk patients. It is important to note that CheckMate 214 was the first combination therapy in the modern era with the most extended follow-up of 55 months, which considers stopping treatment of nivolumab-plus-ipilimumab after two years of administration. Even though the most frequent treatment-related adverse events were various immune-related adverse events, these patients experienced periods of durable disease control after discontinuing ICIs. Moreover, single-agent nivolumab maintenance therapy provided durable response and survival benefits and maintained the quality of life [32,46].

On the other hand, all the IO/TKI combinations have demonstrated superiority over sunitinib, regardless of the IMDC risk group. Although these trials may not have as long a follow-up as IO-IO regimens, most IO/TKI therapies have been shown to improve outcomes with a significantly higher rate of complete responses. Furthermore, in patients with a high symptom burden or rapidly progressive RCC, prompt action is essential to prevent progression. In these cases, researchers prefer IO-TKI drug combinations due to their higher response rates [47,48].

Unfortunately, no head-to-head clinical trials currently support the comparative effectiveness of IO combinations. However, indirect comparisons from systematic reviews and network meta-analyses suggest that the combination of nivolumab-plus-cabozantinib is likely to be the preferred treatment due to its highest overall survival benefit in the intent-to-treat populations (HR, 0.60; 95%CI, 0.40–0.90). On the other hand, pembrolizumab-plus-lenvatinib showed the most significant improvement in progression-free survival across all risk groups (HR, 0.39; 95% CI, 0.32–0.48). Moreover, pembrolizumab-plus-lenvatinib demonstrated the best overall response rate: 71.3% of patients experienced an objective response rate and 18.3% had a complete response rate [49,50,51,52].

The current guidelines establish a straightforward step-wise approach to the management of advanced and metastatic RCC based on risk stratification. However, in most pivotal trials, primary endpoints like overall survival, progression-free survival, or overall response rates were assessed in the intention-to-treat population rather than according to the risk groups. An exception is found in CheckMate 214, which exclusively investigated primary endpoints across intermediate-risk and poor-risk groups [46]. Consequently, relying solely on risk model stratification for making treatment decisions may be deemed unreasonable. 

In conclusion, researchers suggest using IO/TKI drug combinations for patients who have a high symptom burden or rapidly progressive RCC, as they show higher response rates. This combination also has the advantage of not being restricted by risk grouping. However, ICI doublets provide the most long-lasting benefits. It is essential to always take into account patients’ comorbidities and the possibility of adverse reactions to the drugs.

## 3. A Deep Dive into Treatment Particularities in Advanced Renal Cell Carcinoma

Treating mRCC is a complex and highly individualized process, taking into account various factors such as the physician’s experience with each treatment regimen, the patient’s medical history, performance status, comorbidities, and potential adverse effects of the therapy. Despite advancements in cancer biology, managing metastatic disease remains a significant challenge. As previously mentioned, in advanced RCC, the evolving concept of risk stratification plays a pivotal role. The Memorial Sloan Kettering Cancer Centre (MSKCC) and the International Metastatic Renal Cell Database Consortium (IMDC) risk scores play a crucial role in this risk stratification, distinguishing three risk groups to estimate patients’ survival. Both scores serve as composite prognostic biomarkers, incorporating biological and clinical parameters and providing essential information to guide treatment decisions [53]. While phase III trials in metastatic RCC have demonstrated statistical and clinical significance in the overall population, certain features bear relevant clinical implications for mRCC patients. These features include the metastatic burden and the specific sites of metastasis. 

The prognosis of mRCC greatly depends on the sites of metastasis, with the lungs and bones being the most frequently observed distant metastatic sites. Patients with pulmonary metastases often experience the most promising outcomes, with a median overall survival of 25.1 months. In contrast, bone metastases increase the risk of skeletal-related events, significantly impacting mortality and decreasing the quality of life for RCC patients. For example, in a subgroup analysis of over 11,000 metastatic RCC patients, the median overall survival was 19.4 months for patients with bone metastases and 22 months for those with bone and other visceral metastases. Liver metastases occur less frequently than lung or bone metastases and are correlated with poor prognosis, with a median overall survival of 17.6 months (95% CI 16.0–19.2) [54,55,56,57].

On the other hand, brain metastases not only signify a poor prognosis but also exhibit specific responses to oncological treatments. The blood–brain barrier may limit drug delivery to brain tumors. As a result, large molecules such as biological drugs and monoclonal antibodies do not cross this barrier [58,59,60]. However, for example, cabozantinib effectively crosses the blood–brain barrier and has shown clinical and radiographic responses in RCC brain metastases [61,62].

Therefore, consideration of the metastatic sites is important in guiding treatment options for metastatic kidney cancer.

The current recommendations for RCC are derived from traditional randomized controlled trials, with guidelines assessing the strength of their recommendations based on clinical trial results reporting progression-free survival PFS and OS outcomes. However, these clinical guidelines do not take into consideration the specific site of metastasis.

Sunitinib malate, a multitargeted receptor tyrosine kinase inhibitor, has demonstrated both antitumor and antiangiogenic activity. Since its approval in 2006, it has been considered the gold-standard systemic treatment for metastatic RCC. After almost two decades, it remains a valid option for all metastatic RCC risk groups.

However, most IO combinations have replaced sunitinib as the preferred first-line therapy for metastatic clear cell RCC, as per the NCCN guidelines. The recommended regimens include ipilimumab + nivolumab and cabozantinib, with the latter recommended only in poor/intermediate-risk groups. Additionally, axitinib + pembrolizumab, cabozantinib + nivolumab, lenvatinib + pembrolizumab, ipilimumab + nivolumab, and cabozantinib are listed for both favorable and poor/intermediate-risk groups (Figure 1). Other recommended therapies include axitinib + avelumab, pazopanib, or sunitinib [33,46,63,64,65,66,67].

As presented before, all IO-IO or IO-TKI regimens have been compared to sunitinib and consistently demonstrated improved outcomes. In contrast, the Comparz trial showed that pazopanib and sunitinib have similar efficacy. However, in this trial, the safety and quality-of-life profiles favored pazopanib [68]. Table 1 presents the study’s design and the main adverse reactions.

Different metastatic sites may exhibit varying sensitivity to specific treatment regimens. However, even though all trials may have examined the distribution of site-specific metastases in RCC patients, not all of them reported outcomes based on the metastatic sites at the time of cancer diagnosis. Therefore, in this review, we focus on the first category of therapies based on pivotal trials in metastatic RCC and explore the impact of metastatic sites on treatment outcomes (Table 2).

### 3.1. Bone Metastases

Bone metastases concern one-third of patients with mRCC and often lead to complications known as skeletal-related events (SREs). Skeletal morbidity associated with bone metastases includes severe pain, hypercalcemia, impaired mobility, and pathologic fractures that may require surgery or radiotherapy. These complications significantly contribute to morbidity and have a substantial impact on both survival and quality of life. Recent data indicate that over 80% of mRCC patients may experience SREs. However, there is currently limited published evidence regarding the effects of ICIs or TKIs on bone metastases [69,70].

Most importantly, both preclinical and clinical studies have demonstrated that TKIs act on osteoblasts and inhibit osteoclastic bone resorptive activity [71]. Although there are no prospective clinical studies supporting the efficacy of sunitinib in mRCC with bone metastases, retrospective data have shown survival benefits compared to cytokines (24 months versus 18 months; *p* < 0.01), along with reduced bone pain, fractures, and development of new bone metastases. These findings are further supported by another retrospective study conducted by Zolnierek et al., which compared the efficacy of targeted therapy in patients with RCC with pre-existing or new bone metastases. In 292 patients with metastatic RCC, treatment with sunitinib reduced the formation (*p* = 0.034) and time to new bone metastases (*p* = 0.047) compared with sorafenib [72,73,74].

However, cabozantinib, a third-generation TKI, has demonstrated superiority over sunitinib. The significant role of this class of agents is supported by results from the randomized phase II CABOSUN trial, where cabozantinib exhibited a 31% reduction in the median rate of progression or death compared to sunitinib [75]. The promising clinical activity of cabozantinib on mRCC patients with bone metastases has been demonstrated in subgroup analyses from the phase II CABOSUN trial and phase III METEOR trial. In the phase III METEOR study, cabozantinib significantly improved outcomes for patients with bone metastases compared to everolimus. In this subgroup of patients, treatment with cabozantinib nearly doubled progression-free survival (7.4 months with cabozantinib versus 2.7 months with everolimus, HR, 0.33) and overall survival (20.1 months with cabozantinib versus 12.1 months with everolimus, HR, 0.54). Moreover, objective responses were observed in 17% of patients receiving cabozantinib compared to 3%, whereas no patients had a confirmed response with everolimus [66]. 

It is important to note that the data mentioned earlier are derived from studies conducted before the approval of immune checkpoint inhibitors. It is surprising that there is limited information available on the effectiveness and safety of immunotherapy in patients with metastatic RCC who have bone metastases. While a few case reports have shown positive results in terms of radiological response in metastatic RCC patients treated with ICIs, there is still a need for more prospective data [76,77,78,79,80]. Nevertheless, reliable evidence on the efficacy of ICIs in bone metastases primarily comes from a subgroup analysis of the CheckMate 025 trial. This analysis demonstrated superior overall survival with nivolumab (18.5 months, 95% CI = 10.2–not reached), compared to everolimus (13.8 months, 95% CI = 7.0–16.4) [81,82]. 

The efficacy benefits of combining nivolumab with cabozantinib in subgroups of patients with bone metastases are consistent with the advantages previously observed in nivolumab or cabozantinib as monotherapies versus everolimus [66,82]. 

Updated results from the CheckMate 9ER trial establish nivolumab-plus-cabozantinib as a potent primary option for advanced RCC patients with bone metastasis. In 152 patients with bone metastases, the median PFS was 18.2 months (95% CI, 8.3–20.1) in the ICI + TKI arm, compared to 4.4 months (95% CI, 3.7–7.0) with sunitinib monotherapy (HR, 0.38; 95% CI, 0.25–0.59). In those without bone metastasis, the median PFS was 17.0 months (95% CI, 12.5–20.0) with the combination and 9.5 months (95% CI, 7.9–11.0) with sunitinib (HR, 0.57; 95% CI, 0.45–0.72). Patients with bone metastases benefit from nivolumab and cabozantinib combination therapy, resulting in improved outcomes and prolonged survival compared to subgroups without bone metastases [64,83,84].

In the CLEAR trial, another combination immunotherapy demonstrated superiority over the standard of care, sunitinib. With extended four-year follow-up data, lenvatinib-plus-pembrolizumab continues to exhibit superiority to sunitinib, particularly in population subgroups based on the site of metastasis. There was clinically relevant efficacy observed in subgroups of patients with baseline bone metastases. For instance, in patients without bone metastases, the median PFS was 23.4 months in the combination arm and 9.7 months in the sunitinib arm (HR, 0.42; 95% CI 0.33–0.54). In contrast, patients with bone metastases demonstrated a PFS of 24.3 months versus 5.6 months in the lapatinib-plus-pembrolizumab versus sunitinib arms (HR, 0.33; 95% CI 0.21–0.52). These results were higher than in other subgroups with visceral metastases, such as in patients with liver metastases (HR, 0.43; 95% CI 0.25–0.75). Additionally, 22.5% of patients with bone metastases exhibited an HR of 0.50 (95% CI 0.30–0.83) for OS, surpassing patients with baseline lung metastases (HR, 0.57; 95% CI 0.40–0.80) [65,85].

### 3.2. Visceral Metastases

The optimal selection of patients for first-line mRCC is challenging. Unfortunately, no specific biomarker has been identified to aid in selecting the ideal patient for the most appropriate therapy. Currently, treatment algorithms rely on risk stratification. However, it is crucial to consider the symptom burden in cancer patients.

Patients with visceral rapid symptoms require rapid-acting treatment for clinically significant symptom control. Therefore, combining ICI and TKI is most beneficial when the patient needs a prompt response.

On the other hand, if the patient is asymptomatic, other factors such as drug-related toxicity may influence the choice of first-line treatment [86,87,88,89,90,91].

In a post hoc exploration of the CheckMate 9ER trial, a higher proportion of patients experienced tumor shrinkage with nivolumab-plus-cabozantinib compared to sunitinib across all organ sites. In the exploratory assessment regarding the depth of response in target lesion organ sites, a greater proportion of patients exhibited tumor shrinkage with nivolumab-plus-cabozantinib versus sunitinib, irrespective of organ sites. The HR was similar in bone (HR 0.51 (0.40–0.64)) and hepatic metastases (HR 0.51 (0.33–0.79)), but more favorable in lung metastases (HR 0.38 (0.25–0.59)) [83]. Additionally, there were 7 complete responses observed in patients with bone metastases, 5 in those with liver metastasis, and 24 in those with lung metastasis. Further analysis revealed that the confirmed objective response rate was 52% in bone metastases, which was comparable to that of liver metastasis, while it was higher in lung metastasis at 56%. This suggests that the combination of nivolumab and cabozantinib is equally effective in bone and liver metastases but more potent in lung metastases [83].

The CheckMate 214 trial, with a follow-up of over five years, investigated the effectiveness of dual ICIs in treating visceral metastases. The combination of nivolumab and ipilimumab demonstrated promising results in the treatment of bone and visceral metastases. However, the trial revealed a higher HR in the case of lung (HR 0.61, 0.47–0.78) and liver (HR 0.64, 0.42–0.96) metastases compared to bone (HR 0.71, 0.47–1.08) or lymph nodes (HR 0.79, 0.59–1.07) [46].

### 3.3. Brain Metastases

It has been observed that approximately 3% to 17% of patients with advanced kidney cancer develop brain metastases, leading to a poor prognosis [92,93,94,95,96,97]. Unfortunately, the exclusion of patients with intracranial metastases from almost all major clinical trials underscores the urgent need for more research in this area. Brain metastases represent an unmet clinical need that requires further attention and investigation. 

Recent research has challenged the belief that the brain is an immunological refuge protected by the blood–brain barrier. It has been found that other immune cells, including activated lymphocytes CD8+ and regulatory T cells, can migrate across central nervous system barriers and play a crucial role in the occurrence and development of brain metastasis [98,99]. However, brain metastases are associated with a complex tumor microenvironment, characterized by dense infiltration of tumor-infiltrating lymphocytes expressing inhibitory factors of the immune response, such as PD-1 and PD-L1. Hence, there is potential for the use of immunomodulatory drugs in patients with brain metastases and primary CNS tumors.

For a long time, it was believed that the brain was protected by the blood–brain barrier, making it an immunological refuge. However, recent research has shown that besides microglia and perivascular macrophages, other immune cells are present in the central nervous system. These resident immune cells include a small number of activated lymphocytes, such as CD8+ and regulatory T cells, which can cross the central nervous system’s barriers, creating a complex microenvironment in the tumor. These lymphocytes produce PD-1 and PD-L1 proteins, which inhibit the immune response. Therefore, immunotherapy with immune checkpoint inhibitors could be a viable treatment option for patients with primary CNS tumors or brain metastases [100,101].

Unfortunately, the first phase III trial, CheckMate 214, which investigated the efficacy and safety of first-line double immunotherapy with nivolumab-plus-ipilimumab, excluded all patients with central nervous system involvement [46].

CheckMate 920 is a community-based, multi-arm, phase IIIb/IV clinical trial conducted in the USA. It is the first and only trial that evaluates the safety and effectiveness of nivolumab-plus-ipilimumab as a first-line treatment in mRCC patients. These patients exhibit clinical features that are typically excluded from phase III trials, including non-clear cell RCC, brain metastases, or low-performance status. The study enrolled a total of 28 patients in the brain metastases cohort. An interim analysis revealed that all patients exhibited partial responses, and a nine-month progression-free survival was observed with a 95% confidence interval of 2.9 to not estimable. The results from this study provide valuable data for the treatment of patients with RCC and brain metastases, addressing a gap in the existing evidence [102].

The phase II trial GETUG-AFU 26 NIVOREN addressed whether ICI has the potential to improve the prognosis of RCC patients with brain metastasis. The French trial evaluated the safety and efficacy of nivolumab in 83 real-world patients with CNS metastases who had already progressed on prior antivascular endothelial growth factor targeted therapy. Unfortunately, the final results showed a brain metastasis response rate of only 12%. The authors concluded that nivolumab’s activity is limited in intracranial secondary tumors compared to extracranial lesions. They also highlighted the importance of a comprehensive treatment approach in brain metastases that integrates local therapy and systemic treatment strategies [103].

#### 3.3.1. What about TKI in Metastatic Brain RCC?

The blood–brain barrier presents a challenge for the circulation of anticancer drugs to the central nervous system. However, small-molecule tyrosine-kinase inhibitors can cross the blood–brain barrier and exert an effect within the CNS [104]. Specifically, sorafenib, sunitinib, pazopanib, or cabozantinib are considered safe and effective therapies for treating RCC brain metastases [105,106,107,108].

A limited retrospective analysis has explored the impact of tyrosine kinase inhibitors on brain metastases in RCC patients [106,107]. For instance, a retrospective sub-analysis of the phase III TARGET study revealed that the incidence of developing new cerebral secondary lesions in patients treated with sorafenib was lower than in patients receiving a placebo (3% vs. 12%, *p* < 0.05). Therefore, the authors concluded that sorafenib may reduce the occurrence of new brain metastases, even if it does not improve overall survival [105].

Another TKI that demonstrated intracranial clinical benefit is sunitinib. Gore et al. reported in a large global expanded-access trial in metastatic RCC that patients treated with sunitinib exhibited a 42% clinical benefit rate, with stable disease observed in 33% of cases [108]. 

Moreover, several case reports support the use of pazopanib and cabozantinib as potent TKIs for treating RCC patients with cerebral metastasis [109,110,111,112,113]. Notably, Hingorani M. et al. presented a case of pazopanib-induced regression of brain metastasis after whole-brain palliative radiotherapy in an RCC patient [114]. 

In a retrospective cohort study reported in JAMA Oncology, Hirsch et al. support the safety and effectiveness of cabozantinib in treating cerebral metastases by crossing the blood–brain barrier. The study involved data from 88 patients with metastatic RCC and brain metastases treated with cabozantinib in 15 United States and European institutions. A complete intracranial response was noted in 10% of patients, and stable disease was observed in 32% [61].

#### 3.3.2. Finally, Are ICIs + TKI Combinations Active in Treating Brain Metastases from Renal Cell Carcinoma?

Most main clinical trials on combination therapies involving immune checkpoints and tyrosine kinase inhibitors have typically excluded patients with brain metastases. However, a few trials have allowed the inclusion of patients with brain metastases. Notably, in the Javelin 101 phase III study of nivolumab-plus-axitinib, 5.2% of patients had asymptomatic or controlled cerebral metastasis. In this analysis, progression-free survival for patients assigned to combination therapy was higher than in patients receiving sunitinib (4.9 months, 95% CI: 1.6, 5.7 vs. 2.8 months, 95% CI: 2.3, 5.6). During the trial, among patients without brain metastasis at enrolment, secondary brain tumors occurred in eight patients from the combination arm and ten on the sunitinib arm developed brain metastasis [33]. We found no other data on intracranial efficacy using other ICI plus TKIs in the pivotal trials.

After analyzing the evidence, we concluded that combining nivolumab with ipilimumab is the most effective treatment for lung and lymph node metastases. For bone metastases, cabozantinib in monotherapy or cabozantinib-plus-nivolumab are potent options for advanced RCC patients with bone metastasis. Although most patients with brain metastases were excluded from the main trials, limited studies would provide valuable data for the ICI treatments of patients with RCC and brain metastases. Moreover, sorafenib may reduce the occurrence of new brain metastases, although it does not improve overall survival. 

## 4. RCC with Sarcomatoid Features

Sarcomatoid dedifferentiation, a prevalent form of tumor dedifferentiation, is characterized by spindle-shaped components that resemble sarcoma cells. Interestingly, sarcomatoid RCC is not categorized as a unique tumor subtype since it can be observed in any histologic subtype of RCC but is more commonly found in clear cells and chromophobe RCC. This aggressive histologic growth pattern is seen in 5% of RCC patients and 20% of metastatic cases. Due to its aggressive nature, clear-cell or papillary RCC with any proportion of sarcomatoid features would be classified as an ISUP grade four tumor. Therefore, understanding this type of tumor dedifferentiation is crucial in devising effective treatment plans for patients with RCC [115,116,117].

Sarcomatoid dedifferentiation is known to have a weak response to tyrosine kinase inhibitor monotherapy. However, immune checkpoint therapy combinations have shown remarkable responses, with nivolumab-plus-ipilimumab being the best option [118].

An analysis of 139 intermediate or poor-risk patients with sarcomatoid RCC was conducted in a post hoc phase III CheckMate 214 trial. The latest findings from this trial indicate that nivolumab + ipilimumab showed clinically significant benefits in long-term overall survival when compared to sunitinib. The patients who received nivolumab + ipilimumab had an OS of 48.6 months, while the ones who received sunitinib had an OS of 14.2 months. The hazard ratio was 0.46, with a 95% confidence interval of 0.29–0.71. Moreover, according to researchers, the presence of sarcomatoid is the top biomarker to predict response to immune checkpoint therapy [32]. Therefore, it is worth considering this treatment option for better outcomes in patients with metastatic renal cell carcinoma and sarcomatoid dedifferentiation.

## 5. Conclusions and Future Directions

In the current clinical practice, IO-IO and IO-TKI represent the backbone treatments for metastatic RCC. Unlike some other cancers, RCC lacks approved biomarkers to guide therapy. Therefore, understanding which patients and who will benefit more from a specific therapy remains a challenge. 

The applicability of the IMDC risk stratification has decreased, and factors such as tumor burden, number, and locations of metastasis play a crucial role in guiding therapy. Based on higher overall response rates, there is a better chance of controlling the disease with IO-TKI compared to IO-IO. On the other hand, for patients with a low disease burden, responses to IO-IO are durable, even after discontinuing the treatment.

Different sites of metastatic disease can exhibit unique clinical outcomes. For example, cabozantinib has demonstrated effectiveness in treating RCC, showing improvement in bone metastases. Conversely, double ICIs have proven to be a viable therapeutic option in patients with visceral lung and liver secondary lesions. Additionally, when compared to sunitinib, the combination of nivolumab-plus-axitinib has been shown to be more effective in preventing the development of intracranial disease.

For a truly personalized approach, it is essential to understand the side effect profile of each treatment. Patients receiving IO-IO exhibit a greater risk for severe immune-related adverse effects. Therefore, it should be avoided in patients with underlying autoimmune diseases. However, when using the IO/TKI combination, it is challenging to attribute adverse events to one or both drugs.

Understanding the impact of site-specific metastases on the outcomes of patients with metastatic renal cell carcinoma is crucial. The current lack of data on treatment response to different metastases in sites such as the brain, bone, liver, and lung is alarming and calls for immediate attention. Numerous studies have shown that patients’ prognosis greatly depends on the location of metastases. Hence, a better understanding of this subject is essential to improving patient outcomes and enhancing the effectiveness of treatment options.

## Figures and Tables

**Figure 1 biomedicines-12-01111-f001:**
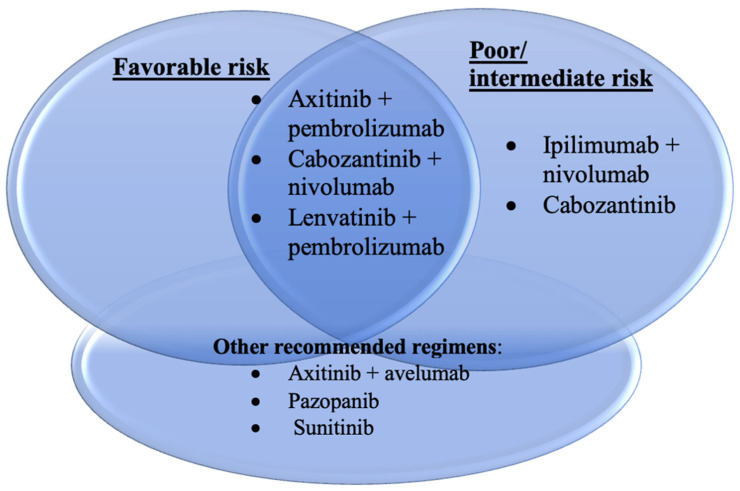
Principles of systemic therapy in metastatic RCC.

**Table 1 biomedicines-12-01111-t001:** Adverse effects in clinical trials.

Trial	Design	Adverse Reactions		
		TRAEs of Any Grade	Grade 3 or 4 Events	TRAEs Leading to Discontinuation
CheckMate 214 - 1096 patients (only intermediate and poor risk) - 1:1 ratio [46]	Nivolumab (3 mg/kg) + ipilimumab (1 mg/kg) IV, q3w, 3 doses, followed by nivolumab q2w	93%	All events: 46% - Increased lipase level 10% - Fatigue 4% - Diarrhea 4%	22%
	Sunitinib (50 mg) PO once daily for 4 weeks (6-week cycle)	97%	All events: 63% - Hypertension 16% - Palmar–plantar erythrodysesthesia 9% - Thrombocytopenia 5%	12%
Javelin Renal 101 - 886 patients - 1:1 ratio [33]	Avelumab (10 mg/kg) IV q2w + axitinib (5 mg) orally twice daily	99.5%	All events: 71.2% - Hypertension 25.6% - Increased alanine aminotransferase level 6% - Palmar–plantar erythrodysesthesia syndrome 5.8%	7.6%
	Sunitinib (50 mg) orally once daily for 4 weeks (6-week cycle)	99.3%	All events: 71.5% - Hypertension 17.1% - Anemia 8.2% - Thrombocytopenia 6.2 - Neutropenia 8% - Decreased platelet count 5%	13.4%
KEYNOTE-426 - 861 patients - 1:1 ratio [63]	Pembrolizumab (200 mg) IV once every 3 weeks + axitinib (5 mg) orally twice daily	98.4%	All events: 75.8% - Hypertension 22.1 - Diarrhea 9.1% - Alanine aminotransferase increased 13.3	30.5%
	Sunitinib (50 mg) orally once daily for the first 4 weeks of each 6-week cycle	99.5%	All events: 70.6% - Hypertension 19.3% - Fatigue 6% - Diarrhea 4.7%	13.9%
CheckMate 9ER - 651 patients - 1:1ratio [64]	Nivolumab (240 mg every 2 weeks) + cabozantinib (40 mg once daily)	96.6%	All events: 75.3% - Hypertension 12.5% - Palmar–plantar erythrodysesthesia 7.5% - Diarrhea 6.9% - Increased lipase level 6.2%	19.7%
	Sunitinib (50 mg once daily for 4 weeks of each 6-week cycle)	93.1%	All events: 70.6% - Palmar–plantar erythrodysesthesia 7.5% - Hypertension 13.1% - Hyponatremia 5.9%	16.9%
CLEAR - 1069 patients - 1:1:1 ratio [65]	Lenvatinib (20 mg orally once daily) + pembrolizumab (200 mg IV q3w)	99.7%	All events: 82.4% - Hypertension 27.6% - Weight decrease 8% - Proteinuria 7.7%	37.2%
	Lenvatinib (18 mg orally once daily) + everolimus (5 mg orally once daily)		All events: 83.1% - Hypertension 22.5% - Diarrhea 11.5% - Fatigue 7.6%	27.0%
	Sunitinib (50 mg orally once daily, alternating 4 weeks receiving treatment and 2 weeks without treatment)	98.5%	All events: 71.8% - Hypertension 18% - Diarrhea 5.3% - Fatigue 4.4%	14.4%
METEOR - 658 patients - 1:1 ratio [66]	Cabozantinib at a dose of 60 mg daily	100%	All events: 68% - Hypertension 15% - Diarrhea 11% - Fatigue 9%	9%
	Everolimus at a dose of 10 mg daily	99%	All events: 58% - Anemia 16% - Fatigue 7 - Hyperglycemia 5%	10%

**Table 2 biomedicines-12-01111-t002:** Summary of the phase III clinical trials for first-line treatment of patients with metastatic RCC.

Phase III Trials	Metastases	No. ≥ 2	Lung	Liver	Bone	Lymph Nodes	Brain
**CheckMate 214 trial** **Nivolumab +** **ipilimumab** [46]	Incidence (%)	335(79%)	381 (69%)	99 (18%)	112 (20%)	264 (45%)	Excluded all patients with brain metastases
	HR(95% CI) vs. sunitinib (PFS)	NR	NR	NR	NR	NR	
	HR (95% CI) vs. sunitinib (OS)	NR	0.61 (0.47–0.78)	0.64 (0.42–0.96)	0.71 (0.47–1.08)	0.79 (0.59–1.07)	
**Javelin Renal 101** **Avelumab + axitinib** [33]	**Patients and methods**	- Randomized phase III trial, 1:1 to receive either avelumab (10 mg/kg intravenously every 2 weeks) plus axitinib (5 mg orally twice daily) or sunitinib (50 mg orally once daily for 4 weeks; 6-week cycle) - Continued treatment until confirmed disease progression, unacceptable toxicity, refusal to participate further, or loss to follow-up					
	Adverse events	- Avelumab-plus-axitinib arm: TEAEs of any grade occurred in 434 (100%), grade ≥3 TEAEs in 81.1% and 31.8% discontinued due to a TEAE - Sunitinib arm TEAEs of any grade occurred in (99.3%, including grade ≥3 TEAEs in 77.4% and 16.2% discontinued due to a TEAE					
	Incidence	254 (57.4%)	332 (76.5%)	83 (18.7%)	97 (21.9%)	196 (44.3%)	Excluded patients with symptomatic or active with brain metastases
	HR (95% CI) vs. sunitinib (PFS)	NR	NR	NR	NR	4.9 mo (95% CI: 1.6, 5.7)	
	HR (95% CI) vs. sunitinib (OS)	NR	NR	NR	NR	NR	
**KEYNOTE-426** **Pembrolizumab + axitinib** [63]	Incidence	315 (73%)	312 (72.2%)	66 (15.3%)	103 (23.8%)	199 (46.1%)	Excluded patients with symptomatic or active with brain metastases
	HR (95% CI) vs. sunitinib (PFS)	NR	NR	NR	NR	NR	
	HR (95% CI) vs. sunitinib (OS)	NR	NR	NR	NR	NR	
**CheckMate 9ER** **Nivolumab + cabozantinib** [64]	Incidence	NR	238 (73.7%)	73 (22.6%)	78 (24.1%)	130 (40.2%)	Excluded patients with symptomatic or active with brain metastases
	HR (95% CI) vs. sunitinib (PFS)	NR	0.51 (0.40–0.64)	0.51 (0.33–0.79)	0.38 (0.25–0.59)	NR	
	HR (95% CI) vs. sunitinib (OS)	NR	0.63; (0.46–0.86)	0.47; (0.27–0.82)	0.64; (0.39–1.06)	NR	
**CLEAR** **Lenvatinib-plus-pembrolizumab** [65]	Incidence	254	249 (70.1%)	60 (16.9%)	85 (23.9%)	NR	Excluded patients with brain metastases
	HR (95% CI) vs. sunitinib (PFS)	NR	0.32 (0.25–0.41)	0.43 (0.25–0.75)	0.33 (0.21–0.52)	NR	
	HR (95% CI) vs. sunitinib (OS)	0.56 (0.40–0.79)	0.57; (0.40–0.80)	0.52; (0.27–0.99)	0.50; (0.30–0.83)	NR	
**METEOR** **Cabozantinib vs. everolimus** [66]	Incidence	269 (40.8%)	204 (31%)	88 (27%)	77 (23%)	206 (62%)	Recruited 3 patients with previously treated brain metastases
	HR (95% CI) vs. sunitinib (PFS)	NR	NR	NR	0.33 (0.21–0.51)	NR	
	HR (95% CI) vs. sunitinib (OS)	NR	NR	NR	0·54 (0.34–0.84)	NR	

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
