# Peer review of "Metastatic Kidney Cancer: Does the Location of the Metastases Matter? Moving towards Personalized Therapy for Metastatic Renal Cell Carcinoma"

_biomedicines, 2024, doi:10.3390/biomedicines12051111_

Round 1

Reviewer 1 Report

Comments and Suggestions for Authors

I think their manuscript is a well-organized on the treatment of metastatic renal cell carcinoma according to the site of metastasis. Starting from the introduction, they discuss current systemic therapy for metastatic renal cell carcinoma, and since the response to treatment and prognosis vary depending on the site of metastasis, they conclude by summarizing the latest information on treatment according to metastatic sites.

However, there are some considerations regarding this paper.

While it's understood that lung metastases have a good prognosis, I believe it would be beneficial to include information on lung metastases as well in this manuscript. Similarly, I think it would be better to include information on lymph node metastases as much as possible. There are reports that Nivo+Ipi is less effective for lymph node metastases.

Many cases of mRCC have multiple metastatic sites. They summarized each treatment with examples of PFS and OS, making it difficult to assess the effectiveness for each metastatic site. It is more important to know how much each metastatic site shrinks with each treatment. While I understand that there is limited evidence on this matter, I believe the paper would be improved by adding information on the extent of treatment effectiveness for each metastatic site, even if it comes from retrospective studies or reports of few cases.  

There is no mention of histology. For example, it is well known that Nivo+Ipi is effective in cases involving a sarcomatoid component, but there is little mention of histological subtypes in this paper.

Author Response

We appreciate your feedback and suggestions on our manuscript. Based on your comments, we have made appropriate revisions to the paper. Thank you for your time and input.

We thank the reviewer again for the close reading of our article and all the constructive and detailed comments.

Comment 1: While it's understood that lung metastases have a good prognosis, I believe it would be beneficial to include information on lung metastases as well in this manuscript. Similarly, I think it would be better to include information on lymph node metastases as much as possible. There are reports that Nivo+Ipi is less effective for lymph node metastases.

Answer: We thank the reviewer for raising this important issue. We have included information on lung and lymph node metastases in the subchapter on visceral metastases.

It is well known that RCC most frequently metastasizes to the lungs (50–60%), and lung metastases have the best prognosis. Data from the pivotal trials showed that most therapeutic combinations, such as nivolumab and cabozantinib or nivolumab plus ipilimumab, are more potent in lung metastases. However, double ICI is less efficient in lymph node metastases.

Comment 2: Many cases of mRCC have multiple metastatic sites. They summarized each treatment with examples of PFS and OS, making it difficult to assess the effectiveness for each metastatic site. It is more important to know how much each metastatic site shrinks with each treatment. While I understand that there is limited evidence on this matter, I believe the paper would be improved by adding information on the extent of treatment effectiveness for each metastatic site, even if it comes from retrospective studies or reports of few cases.

Answer: We appreciate the reviewer's feedback, and we plan to explore this idea in future articles. Unfortunately, we only reviewed the main trials. However, we added a new table to define treatment-specific adverse reactions better.

Comment 3: There is no mention of histology. For example, it is well known that Nivo+Ipi is effective in cases involving a sarcomatoid component, but there is little mention of histological subtypes in this paper.

Answer: We are very grateful to the reviewer for the reminder provided. We have added a new chapter to address the reviewer's question regarding the sarcomatoid component.

  1. RCC with sarcomatoid features

Sarcomatoid dedifferentiation, a prevalent form of tumor dedifferentiation, is characterized by spindle-shaped components that resemble sarcoma cells. Interestingly, sarcomatoid RCC is not categorized as a unique tumour subtype since it can be observed in any histologic subtype of RCC but is more commonly found in clear cells and chromophobe RCC. This aggressive histologic growth pattern is seen in 5% of RCC patients and 20% of metastatic cases. Due to its aggressive nature, clear-cell or papillary RCC with any proportion of sarcomatoid features would be classified as an ISUP grade 4 tumor. Therefore, understanding this type of tumor dedifferentiation is crucial in devising effective treatment plans for patients with RCC [116-118].

Sarcomatoid dedifferentiation is known to have a weak response to tyrosine kinase inhibitor monotherapy. However, immune checkpoint therapy combinations have shown remarkable responses, with nivolumab plus ipilimumab being the best option [119].

An analysis of 139 intermediate or poor-risk patients with sarcomatoid RCC was conducted in a post hoc phase III CheckMate 214 trial. The latest findings from this trial indicate that nivolumab + ipilimumab showed clinically significant benefits in long-term overall survival when compared to sunitinib. The patients who received nivolumab + ipilimumab had an OS of 48.6 months, while the ones who received sunitinib had an OS of 14.2 months. The hazard ratio was 0.46, with a 95% confidence interval of 0.29–0.71. Moreover, according to researchers, the presence of sarcomatoid is the top biomarker to predict response to immune checkpoint therapy [32]. Therefore, it is worth considering this treatment option for better outcomes in patients with metastatic renal cell carcinoma and sarcomatoid dedifferentiation.

We sincerely thank the reviewer again for providing constructive feedback to improve our manuscript!

Reviewer 2 Report

Comments and Suggestions for Authors

In the manuscript entitled “Metastatic kidney cancer: Does the location of the metastases matter? Moving towards personalised therapy for metastatic renal cell carcinoma”, the authors provide evidence regarding questions pertaining to metastatic sites in treatment decisions of renal cell carcinoma (RCC) and the selection of immunotherapy in conjunction with tyrosine kinase inhibitors. Answering these questions expands our knowledge of personalized medicine in RCC management. In this review, the authors list prior studies on trends of systemic therapy against metastatic RCC, and treatment particularities in advanced RCC, including bone, visceral, and brain metastases.  The authors provided one interesting figure that summarizes the principles of systemic therapy in metastatic RCC. Moreover, they provided one table that provides a summary of the clinical trials in phase III for first-line treatment of patients with metastatic RCC. From this discussion, the authors suggest that metastatic sites of RCC respond differently to therapeutic regimens. There is evidence that capazantinib is effective in treating RCC, particularly in improving bone metastasis outcomes. On the other hand, dual immune checkpoint inhibitors (ICIs) may be an applicable therapeutic option in patients suffering from secondary lesions of the visceral lungs and liver. Additionally, when combined with sunitinib, nivolumab plus axitinib has revealed better effectiveness in preventing intracranial disease than sunitinib on its own.

The current findings are interesting, and the review is clearly written.

Comments: 

1) The provided sections read like narration for the evidence of discussed points without critical aspects/reflection points. At the end of each section, a take-home message is advised to be provided.

2) In Table 1, additional data about the clinical trial should be provided including the used doses, sample/design, adverse events (if any), and retention in the trial.

3) The work lacks future directions that include limitations and what is the next step to translate these findings to clinical settings.

4) The authors are advised to add a separate section on the reported adverse effects of IO/IO or IO/TKI therapeutic regimen.

5) In line 409, the “Conclusions” sections should be replaced by “Conclusions and future directions”.

6) The authors are advised to make the table captions stand-alone. To this end, authors are advised to provide a list of abbreviations describing the full names of all the listed abbreviations in the table.

7) What do the authors specifically mean by “93% of patients receiving nivolumab+ipilimumab experienced any grade of immune-related adverse reactions (irAE)” in lines 92-93. Please, clarify.

8) In line 56, the Tab space should be removed from the section title. Please, address this point in the entire manuscript.

9) In line 87, the Tab space should be removed from the subsection title. To avoid readers’ confusion, please, add numbering to the subsection (2.1.). This point needs to be addressed in the entire manuscript.

10) Please, fix typos in the manuscript. For example:
A) In the title, please correct “personalised” to “personalized”.

B) In line 217, please correct “Limph nodes” to “lymph nodes”.

C) In line 319, please use a consistent font size for the numbers.

11) In line 439, please use a specific Data Availability Statement. 

Comments on the Quality of English Language

Minor editing of the English language is required.

Author Response

We appreciate your feedback and suggestions on our manuscript. Based on your comments, we have made appropriate revisions to the paper. Thank you for your time and input.

Comment 1: The provided sections read like narration for the evidence of discussed points without critical aspects/reflection points. At the end of each section, a take-home message is advised to be provided.

Answer: We thank the reviewer for the constructive remarks. We have revised it accordingly.

Comment 2: In Table 1, additional data about the clinical trial should be provided including the used doses, sample/design, adverse events (if any), and retention in the trial.

and

Comment 4: The authors are advised to add a separate section on the reported adverse effects of IO/IO or IO/TKI therapeutic regimen.

Answer: We greatly appreciate the reviewer for bringing this to our attention. We included a new table including sample/design and adverse reactions for each study

Comment 3: The work lacks future directions that include limitations and what is the next step to translate these findings to clinical settings.

Answer: We appreciate the reviewer for bringing this to our attention. We have revised our conclusions and future directions accordingly.

Comment 5: In line 409, the “Conclusions” sections should be replaced by “Conclusions and future directions”.

Answer: We thank the reviewer for the constructive remarks. We have revised it accordingly.

Comment 6: The authors are advised to make the table captions stand-alone. To this end, authors are advised to provide a list of abbreviations describing the full names of all the listed abbreviations in the table.

Answer: We express our gratitude to the reviewer for providing us with constructive feedback. We have made the necessary revisions to our work based on the comments.

Comment 7: What do the authors specifically mean by “93% of patients receiving nivolumab+ipilimumab experienced any grade of immune-related adverse reactions (irAE)” in lines 92-93. Please, clarify.

Answer: We have carefully considered these suggestions and made the necessary revisions to our work accordingly.

In the CheckMate 214 clinical trial, 93% of patients who were treated with nivolumab and ipilimumab experienced some form of immune-related adverse reactions. These adverse reactions included endocrine, pulmonary, hepatic, renal, digestive, or skin issues, and 46% of these cases were considered grade 3 or 4 events. Additionally, up to 35% of patients required high doses of glucocorticoids to treat these immune-related adverse reactions. [32].

Comment 8: In line 56, the Tab space should be removed from the section title. Please, address this point in the entire manuscript.

Answer: We thank the reviewer for the constructive remarks. We have revised it accordingly.

Comment 9: In line 87, the Tab space should be removed from the subsection title. To avoid readers’ confusion, please, add numbering to the subsection (2.1.). This point needs to be addressed in the entire manuscript.

Answer: We thank the reviewer for the constructive remarks. We have revised it accordingly.

Comment 10: Please, fix typos in the manuscript. For example: A) In the title, please correct “personalised” to “personalized”.

  1. B) In line 217, please correct “Limph nodes” to “lymph nodes”.
  2. C) In line 319, please use a consistent font size for the numbers.

Answer: We thank the reviewer for the constructive remarks. We have revised it accordingly.

Comment 11: In line 439, please use a specific Data Availability Statement.

Answer: We thank the reviewer for the constructive remarks. We choosed to exclude this statement.

Thank you again for your thorough and insightful review. We hope that the revised manuscript meets your satisfaction and look forward to hearing back from you!

Round 2

Reviewer 1 Report

Comments and Suggestions for Authors

The manuscript was properly revised and well written.

There are no particular issues.

Reviewer 2 Report

Comments and Suggestions for Authors

The authors have adequately addressed the raised comments; thanks.